# Number Bias in Clinicians’ Documentation of Actinic Keratosis Removal

**DOI:** 10.3390/jcm13010202

**Published:** 2023-12-29

**Authors:** Phillip G. Holovach, Wei-Wen Hsu, Alan B. Fleischer

**Affiliations:** 1Medical School, University of Cincinnati College of Medicine, Cincinnati, OH 45267, USA; 2Division of Biostatistics and Bioinformatics, University of Cincinnati College of Medicine, Cincinnati, OH 45267, USA; hsuwe@ucmail.uc.edu; 3Department of Dermatology, University of Cincinnati College of Medicine, Cincinnati, OH 45267, USA; fleiscab@ucmail.uc.edu

**Keywords:** radiation effects, statistics and numerical data, implicit bias, outcome assessment, health care, skin diseases, dermatology

## Abstract

Background: Actinic keratosis (AK) is a pre-cancerous skin condition caused by sun exposure. Number bias, a phenomenon that occurs when meaning other than numerical value is associated with numbers, may influence the reporting of AK removal. The present study aims to determine if number bias is affecting healthcare providers’ documentation of patient-provider encounters. Methods: A single-center retrospective chart review of 1415 patients’ charts was conducted at the University of Cincinnati Medical Center. To determine if there was a significant difference between even and odd-numbered AK removals reported, an exact binomial test was used. The frequency of removals per encounter was fitted to a zero-truncated negative binomial distribution to predict the number of removals expected. All data were analyzed with RStudio. Results: There were 741 odd and 549 even encounters. Odd removals were reported at a significantly greater frequency than even *p* < 0.001. Age may be contributing to the observed number bias (*p* < 0.001). One, two, and eight were reportedly removed more frequently, while nine, 13, and 14 were reportedly removed less frequently than expected, respectively. Conclusion: Number bias may be affecting clinicians’ documentation of AK removal and should be investigated in other clinical settings.

## 1. Introduction

Actinic keratosis (AK) is a pre-cancerous lesion of the skin caused by chronic, unprotected sun exposure; it is one of the most common reasons for consultation with a dermatologist in the United States [1,2,3,4]. Despite their prevalence, there is a paucity of basic epidemiologic information on AKs that has only recently begun to be addressed [5]. If left untreated, these lesions can progress to malignancy, with an estimated 0.1–20% of AKs progressing to squamous cell carcinoma (SCC) [4,6]. 

Number bias is a phenomenon that occurs when meaning other than numerical value is associated with a number or category of numbers. Examples of number bias, while not well studied in clinical medicine, have been investigated in the fields of psychology and sociology. Number bias appears in various forms and is prevalent throughout many domains.

In Mandarin Chinese, for example, the number “four” has a similar pronunciation as “death”, while the number “eight” sounds like a word for “wealth.” Research has shown that in China, hotel rooms with “eight” in the price received higher ratings, and stock traders decided more quickly and were more likely to purchase lots ending in the number “eight” [7,8,9]. The association of these numbers with their connotational homophones is similar to the respective fame and infamy of lucky “seven” and unlucky “13” in Western culture. According to a poll with over 30,000 responses, when asked to specify their favorite number, “seven” was submitted by the most participants. Moreover, the number “13” is very poorly received in Western culture, and bias towards this number continues to impact modern infrastructure and architecture—out of 629 condominiums in Manhattan with 13 or more floors; 91% of buildings renamed floor 13 to an alternative or skipped the number entirely [10].

Small and large number bias may be modified by the physical positioning of the person—numerical and spatial cognition involve the same neural circuitry within the parietal lobe [11]. When asked to generate random numbers, study participants preferentially generated sequences of smaller numbers when turning their heads to the left and larger numbers when turning their heads to the right [12]. Literature has shown that humans perceive and think about odd and even numbers differently; our brains take longer to react to and process odd numbers [13]. It is hypothesized that this is due to the linguistic markedness and multiple definitions of the word “odd”, potentially causing people to view even numbers as the norm [14].

While inadequately investigated, number bias has also been noted in the healthcare setting. Within an intensive care unit, a study was conducted to investigate number preference regarding invasive mechanical ventilatory therapy. The number settings on a ventilator for positive-end expiratory pressure, respiratory rate, and inspiratory pressure were all analyzed; patients spent a longer amount of time with odd-numbered settings for all three categories [15]. More broadly, number bias within the domain of age may affect the entire field of clinical research [16]. Arbitrary age cutoffs have been noted even within major medical journals. For example, age cutoffs may allow a 75-year-old person to participate in a study but exclude a 76-year-old without scientific justification, potentially excluding older populations from benefiting from clinical research [17]. 

As evidenced in many fields, number bias may influence physician reporting of AK removal. The present study aimed to determine if number bias influences the documentation of AK removal by analyzing if (a) there is a significant difference between odd and even AK removals per patient-provider encounter or (b) if there exists a preference for, or aversion toward, any specific numbers. The secondary objectives sought to investigate if a significant difference in the number of AKs removed existed between providers of opposite sexes and differences in training: doctor of allopathic/osteopathic medicine (MD/DO) and nurse practitioner/physician’s assistant (NP/PA).

## 2. Materials and Methods

### 2.1. Patient Selection

A single-center retrospective chart review of 1415 patients was conducted at the University of Cincinnati Medical Center (UCMC) for patients who were at least 18 years old and had one or more AKs removed between 2012 and 2021. Although it was not part of our inclusion criteria, all AK removals in this study were performed via liquid nitrogen cryotherapy. Only the most recent encounter was considered for patients who had multiple physician encounters resulting in an AK removal.

A list of patients was generated for this project using CPT codes 17000 (first AK removed), 17003 (two to 14 AKs removed), and 17004 (15 or more AKs removed). Patient age at time of appointment, biological sex, race, ethnicity, number of AKs removed, and healthcare provider were collected from each patient chart. 

### 2.2. Statistical Analysis

Patient encounters were organized into groups based on how many AKs were removed, creating categories “one,” “two,” … through “14”, as CPT codes did not allow for enumeration above 14. The data were assessed as a collective across all healthcare providers included in this study. The number of AK removals was reported, and corresponding demographic information was analyzed using R (version 4.1.3, R Core Team, Vienna, Austria).

The patient–physician encounters were then categorized for multiple independent analyses. An exact binomial test was run to determine if there was a significant difference in the number of patient encounters between even and odd amounts of AKs removed per encounter. A logistic regression model was then used to determine if there was a significant difference in the distribution of even versus odd numbers of Aks removed when considering age of patient, sex of provider, and training of provider MD/DO versus NP/PA.

The results observed were compared with what was expected based on a zero-truncated negative binomial regression model. The model predicted the most common number of Aks destroyed per encounter would be one, with progressively decreasing frequency as the number of removals per encounter increased. The enumerated data were then fitted to a zero-truncated negative binomial regression model to determine if the variables age of patient, sex of provider, or training of provider influenced the distribution. For each category of AK removal, the number of removals per grouping was converted from a total count to a probability by dividing the number of AKs removed in each category by the total number of patient encounters to calculate the probability of having a specific number of AKs removed. Confidence intervals (CI) of 95% were then constructed around the expected number of removals predicted by the model. If the observed data fell outside of the confidence intervals, they were deemed a significant deviation from what was predicted by the zero-truncated negative binomial model. Significance was defined in this way as there are limited statistical tests to analyze single-variable ‘count’ data. All statistical analyses were performed with R (ver. 4.1.3). The significance was set at *p* < 0.05.

Further statistical analysis was conducted within subsets of the data to investigate the effects of age, more specifically on the number of AKs removed per encounter. A nonparametric Kruskal–Wallis test was used to determine if patients who had 13 or 14 AKs removed were significantly different in age when compared to the cohort of patients with fewer removals. Similarly, a nonparametric Kruskal–Wallis test was used to determine if the age of patients who had one or two AKs removed was significantly different than the other patients who had more removals. A nonparametric Wilcoxon rank sum test was then performed to determine if a significant difference in age existed between encounters with eight and nine removals per encounter. The significance was set at *p* < 0.05. 

## 3. Results

This study included 1290 patients (Figure 1). Of these, 774 (60%) were male and 516 (40%) were female. The mean ± standard deviation age at encounter was 69.4 ± 11.0 years. Furthermore, 1289 (99.9%) participants were white, with one (0.1%) being Asian and 0 (0%) of patients being African American, American Indian, Alaskan native, or other (Table 1).

### 3.1. Even vs. Odd

Within the set of encounters, 549 (42.6%) resulted in an even number of AKs being removed, while there were 741 (57.4%) odd-numbered removals. The probability of Aks being removed being an odd number was 0.57 (CI 0.547–0.602). The exact binomial test determined that there was a significantly greater number of odd compared to even removals (*p* < 0.0001).

A logistic regression was run to determine if patient age at the time of encounter, sex of the provider, or training of the provider affected the distribution of even and odd frequencies of AK removals (Table 2). A table for even-numbered removals was not included, as the coefficient estimates would have been the same magnitude as odd removals but with the opposite sign (negative or positive). None of these factors significantly influenced the frequency of even or odd AKs removals (age of patient, *p* = 0.06; sex of provider, *p* = 0.39; training of provider, *p* = 0.36).

### 3.2. Distribution of AK Removals

The distribution of all AKs removed was then fit to a zero-truncated negative binomial distribution (which can accommodate the outcome range from 0 to 14), and 95% confidence intervals were constructed using the normal approximation method. The number of patients who had 1, 2, and 8 AKs removed was higher than expected, while those who had 9, 13, or 14 removed were lower than expected relative to the constructed confidence interval (Figure 2). Age of the patient was found to be a significant factor (*p* < 0.01) influencing the number of AKs removed, while sex of the provider (*p* = 0.59) and training of the provider (*p* = 0.12) were not (Table 3).

The mean patient age from patient encounters (13 and 14 removals) was then compared to the rest of the cohort using a nonparametric Kruskal–Wallis test (Table 4). The test determined that there was not a significant difference in age between those who had 13 or 14 AKs removed when compared to the rest of the group (*p* = 0.28). The effect of age on encounters with one or two removals when compared to encounters with more removals was then investigated using a nonparametric Kruskal–Wallis test (Table 5). The test determined there is a significant difference in age when comparing the mean patient age from encounters with one or two removals to the rest of the cohort (*p* < 0.001). A subset of the data, including encounters with eight and nine removals, was analyzed with a nonparametric Wilcoxon rank sum test (Table 6). The analysis concluded there was not a significant difference in mean patient age between encounters with eight and nine removals (*p* = 0.20).

## 4. Discussion

To our knowledge, this is the first study investigating number bias within the field of dermatology. This study found that healthcare providers preferentially selected odd numbers when reporting AK removal. However, these results may be skewed by the large portion of singular AK removals (349 encounters resulted in 1 AK being removed). Although age, sex of provider, and training of provider were not significant in the logistic regression analysis for even and odd-numbered AK removals, there may be unobserved factors that do influence the distribution of even and odd numbers of AKs removed. One possible explanation for the unexpected distribution of odd removals is that most patients consult with dermatology when they first notice an AK, rather than waiting for multiple lesions to present. This explanation could also be the reason one and two AKs were reportedly removed more than expected and fell outside of the 95% CI. Moreover, the Kruskal–Wallis test investigating one and two removals relative to the rest of the cohort demonstrated a significant difference in age, which supports this explanation.

The zero-truncated negative binomial model determined that patient age may influence the frequency of Aks removed per encounter, but sex and provider training did not. This finding supports current literature, as there is an increased likelihood of developing AKs with older age [1]. This may explain the deviation from the zero-truncated negative binomial distribution.

While the removal of eight AKs occurred more frequently than expected, it was most likely unrelated to the phenomena within spoken Mandarin Chinese, as the study took place at a single midwestern American hospital system. The Wilcoxon rank sum test demonstrated that there was no significant age difference between patients who had eight and nine removals reported, which does not provide an explanation for this observation. The removal of nine AKs was reported to be less than expected, and this could be due to the human tendency to favor round numbers (numbers that end with ‘0’) [18]. However, 10 removals were not reported at a greater than expected frequency. In Japanese culture, the number ‘9’ is a homonym for the word ‘suffering’ and is sometimes avoided [19]. While this is not likely the explanation for the results seen here, avoidance of the number ‘9’ is present in Western culture. An example of this phenomenon is seen in the American company Apple skipping the ‘iPhone 9’ and succeeding the eighth edition of the iPhone with the tenth [20]. Microsoft similarly skipped the ninth edition of their operating system ‘Windows’ and followed the eighth edition with the tenth [21].

It is plausible that 13 removals were reported less frequently than expected due to Western society’s negative association with the number. Fourteen removals may have been reported less than expected due to rounding (up to 15) and the higher reimbursement associated with the billing code for the grouping of 15 or greater removals. Interestingly, the Kruskal–Wallis test determined that patients who had 13 and 14 removals documented did not have a significantly different age from the rest of the cohort. This may be explained by some individuals delaying treatment or increased unprotected sun exposure due to a multitude of factors (e.g., occupation). It is important to identify and address number bias in the clinical setting to optimize patient outcomes and ensure patients are not being overcharged for services. Further research is necessary to unravel the subtle ways number bias may influence clinical medicine. 

While number bias may be affecting clinicians’ documentation of AK removals, it is important to keep in mind that counting and documenting AKs is an imperfect science. The ability to enumerate AKs rests on the subjective clinical diagnosis that can be muddled by the non-uniform morphology, continuousness, and clustering of lesions [22]. It has been shown in the literature that independent AK counting by even experienced dermatologists results in variable and sometimes unreliable counts [23]. Previous research has demonstrated that when a provider is able to spend more time with the patient and implement lesion detailing such as measuring lesion diameter, documenting lesions by body part, and defining the body surface affected by AKs, counts result in a more accurate representation of lesion burden [24,25].

This study had some notable limitations. The data used to investigate number bias in this study were reported numbers from the billing section of patient charts; the number of AKs removed versus reported may vary. Thus, the study results directly reflect the influence of number bias on reporting, not necessarily clinical decision-making itself. While unintentional, our study only included AK removals via cryotherapy. Many providers are equipped with a variety of methods to remove Aks, such as topical immune therapy, surgical curettement, laser therapy, and, for clusters of lesions, field therapy. It is likely that count data would vary depending on the method of removal due to the time spent examining the lesions. However, it is unclear to what extent the effects of number bias may fluctuate depending on the method of AK removal since it is an implicit phenomenon and not an inaccuracy in examination. Another potential limitation of this study was the use of single variable ‘count’ data to compare the frequency of the number of AK removals. There are limited statistical tests to determine significance using this type of data, so significance was defined as a removal frequency that fell outside of the confidence interval predicted by our model, rather than the generation of a *p*-value. Furthermore, there are limited covariates and predictors in the raw dataset, which may make the model interpretation somewhat limited, especially for the patients with one and two AKs removed. Finally, this study occurred at a single center, tertiary care, in an urban hospital system, so the patient population may not be representative of all those at risk for developing AKs. Actinic keratosis removal and its documentation are at the discretion of the treating physician. The number bias in our study may very well be harmless, as patients often return to the office for follow-up removal of remaining Aks, but it could have more severe consequences in other settings.

## 5. Conclusions

Odd numbers were preferentially selected in the reporting of AK removal. Patient age at encounter was found to influence the distribution of the number of AKs removed, with older patients having more AKs removed. Additionally, 1, 2, and 8 AKs were removed more frequently than expected per patient encounter, while 9, 13, and 14 were removed less frequently than expected. Further research should be conducted to elucidate the phenomenon of number bias in more acute settings—such as the ICU—where it may detrimentally affect outcomes and more broadly in clinical research to ensure perceptions of patient age do not incorrectly determine eligibility to participate in studies.

## Figures and Tables

**Figure 1 jcm-13-00202-f001:**
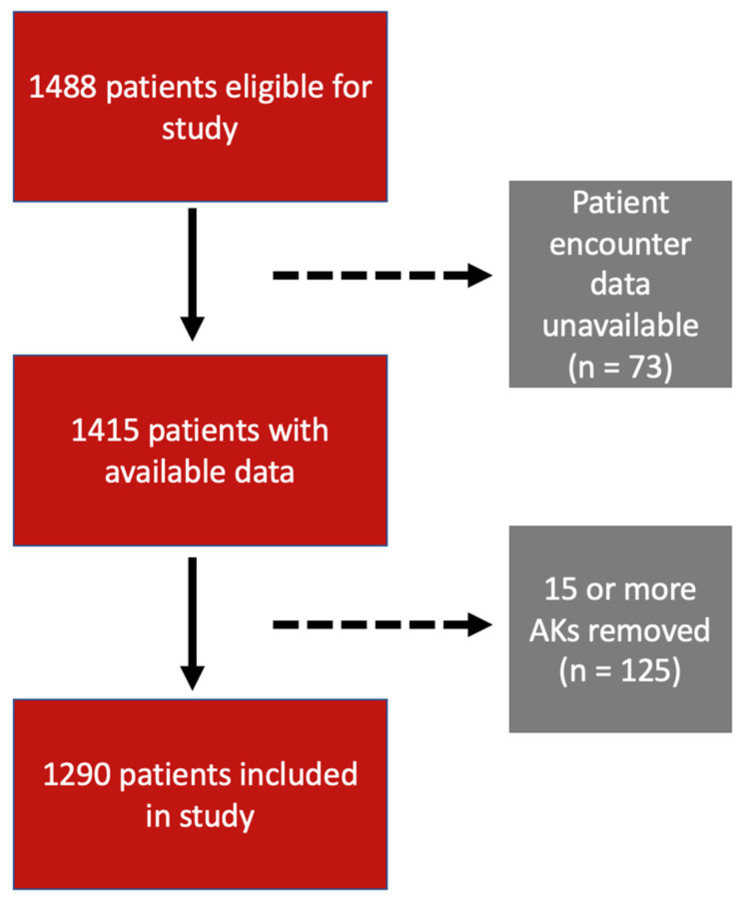
Patient selection flowchart based on inclusion and exclusion criteria.

**Figure 2 jcm-13-00202-f002:**
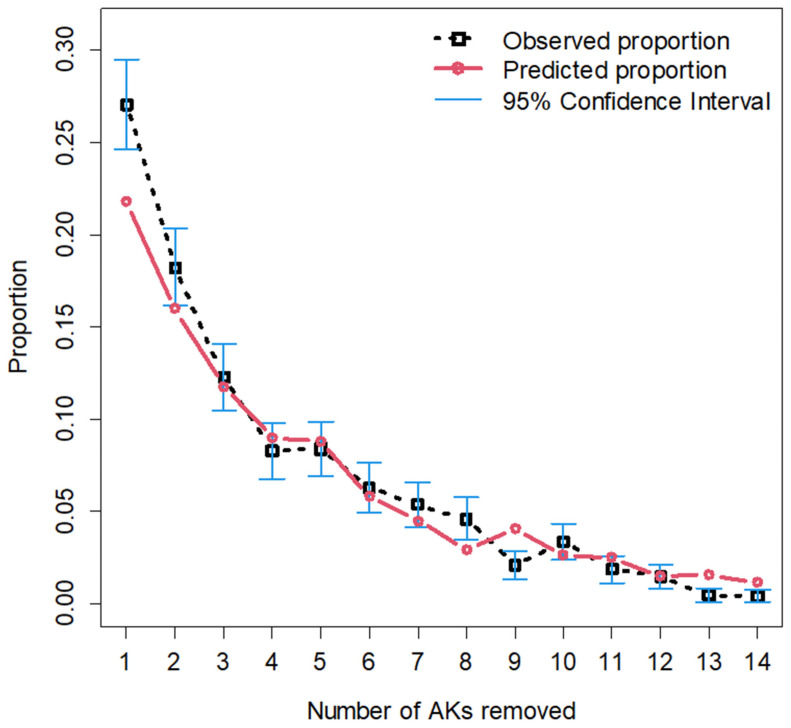
Zero-truncated negative binomial distribution with 95% confidence intervals.

**Table 1 jcm-13-00202-t001:** Characteristics of patients.

Demographic Characteristics	
Number of Patients	1290 (100%)
Age at encounter (years)	69.4 ± 11.0
Sex	
Male	774 (60%)
Female	516 (40%)
Race	
White	1289 (99.9%)
African American	0 (0%)
American Indian or Alaskan Native	0 (0%)
Asian	1 (0.1%)
Other	0 (0%)
Ethnicity	
Hispanic	1 (0.1%)
Non-Hispanic	1289 (99.9%)

Values are presented as numbers (%) or mean ± SD; y, year.

**Table 2 jcm-13-00202-t002:** Estimates of the logistic regression model for odd-numbered AK removals.

Variable	Estimate (S.E.)	*p*-Value
Age	−0.003 (0.002)	0.064
Sex of Provider	0.035 (0.041)	0.390
Training of Provider	−0.032 (0.036)	0.365

S.E.: standard error.

**Table 3 jcm-13-00202-t003:** Estimates of zero-truncated negative binomial model for the distribution of AK removals.

Variable	Estimate (S.E.)	*p*-Value
Age	0.018 (0.003)	<0.0001
Sex of Provider	0.044 (0.081)	0.587
Training of Provider	−0.111 (0.071)	0.118

S.E.: standard error.

**Table 4 jcm-13-00202-t004:** Kruskal–Wallis test comparing 13 and 14 removal encounters with all others.

AKs Removed Per Encounter	Number of Encounters	Mean Age (Standard Deviation)
13	6	76 (8)
14	5	66.8 (16.7)
Others	1279	69.4 (11.0)

Kruskal–Wallis chi-squared = 2.56, *p*-value = 0.28.

**Table 5 jcm-13-00202-t005:** Kruskal–Wallis test comparing one and two removal encounters with all others.

AKs Removed Per Encounter	Number of Encounters	Mean Age (Standard Deviation)
1	349	66.3 (11.4)
2	235	68.0 (11.1)
Others	548	71.4 (10.5)

Kruskal–Wallis chi-squared = 44.79, *p*-value < 0.001.

**Table 6 jcm-13-00202-t006:** Wilcoxon rank sum test comparing encounters with eight and nine removals.

AKs Removed Per Encounter	Number of Encounters Observed	Observed Proportion	Predicted Proportion	Mean Age (Standard Deviation)
8	59	0.046	0.029	73.4 (11.8)
9	27	0.021	0.040	71.3 (10.2)

Wilcoxon rank sum test = 934.5, *p*-value = 0.20.

## Data Availability

The data for this study are unavailable to the public due to the protected health information guidelines outlined in the waiver for informed consent granted through the institutional review board.

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
