# Peer review of "Number Bias in Clinicians’ Documentation of Actinic Keratosis Removal"

_jcm, 2023, doi:10.3390/jcm13010202_

Round 1
Reviewer 1 Report
Comments and Suggestions for Authors
Dear authors,
You have performed an interesting study.
Please address the methods used to remove the AKs. Was it cryo, shave, or topical therapy? The reader should know what is meant by “removal”. In clinical practice, counting the lesions if they are confluent is often difficult. Often, in these cases, field treatment is used.
On the contrary, if surgical removal is chosen, it will probably be a single lesion. So, the removal method could also influence the number of lesions removed. Moreover, if the lesion shows any suspicious signs, it might be removed to exclude SCC in situ or minimally invasive type. In this case, it would be a single lesion removal.
Additionally, as shown by multiple studies, please address the variability and difficulties in assessing AK numbers. For instance, https://www.medicaljournals.se/acta/content/html/10.2340/00015555-2040
Or
https://jamanetwork.com/journals/jamadermatology/fullarticle/478455
Author Response
Dear reviewer,
Attached is our response to your comments.
Thank you for your time.

Reviewer 2 Report
Comments and Suggestions for Authors
In this article, Holovach et al determined if number bias is affecting healthcare providers’ documentation of patient-provider encounters. 1415 patients’ charts was reviewed at the University of Cincinnati Medical Center. There were 741 odd and 549 even encounters. Odd removals were reported at a significantly greater frequency than even. Age may be contributing towards the observed number bias. One, two, and eight were reportedly removed more frequently while nine, 13, and 14 were reportedly removed less frequently than expected, respectively. They concluded that number bias may be affecting clinician’s documentation of AK removal and should be investigated in other clinical settings. This is a very interesting study from a very interesting perspective. I have several questions as follows.
major concerns)
1) The graph in Figure 2 is difficult to understand, so please make it more clear. For example, show the observed population as ◻︎ with a dotted line to make it easier to understand the difference between the observed population and the predicted population.
2) Am I correct in understanding that the removal of AK includes various types of healing, such as healing with imiquimod cream, surgical healing, and healing with liquid nitrogen?
3) Those who have overall removals are more influenced by age. If so, is it possible that those who had 13 or 14 removed are lower than the predicted model because of age? For example, it is possible that those with more AKs are inevitably older and therefore had fewer removed, etc. Can you examine the effect of age in 13 and 14 by comparing 13 and 14 with the others, rather than the effect of age overall? So, if the people were older in 13 and 14 compared to the others, the effect of age would be greater than the effect of number.
4) Is the higher than predicted number of AKs in 1 and 2 due to younger age? It is possible that in younger age groups, AKs are more likely to be detected at a lower stage, such as one or two, and therefore more likely to be removed, e.g., by surgery. To investigate this, you could make a comparison between 1 or 2 and the others, and if 1 or 2 were younger, the effect of age would be greater than the effect of the number of AKs.
5) The difference between 8 and 9 is difficult to explain by 2) or 3) above. Therefore, it is likely that the number 9 is being avoided. Can you find out if there is a difference in age between 8 and 9, if there is no significant difference in age between 8 and 9, it is possible that you are avoiding the number itself.
Author Response

(The authors gave the same response as above.)

Round 2
Reviewer 2 Report
Comments and Suggestions for Authors
The authors replied to our comments appropriately. No additional comments.